# Erythema Nodosum following *Nocardia* Infection: A Case Report

**DOI:** 10.3390/medicina58121873

**Published:** 2022-12-19

**Authors:** Sujing Li, Bingzhou Ji, Yan Teng, Hui Tang, Hong Cui, Xiaohua Tao, Yibin Fan, Youming Huang

**Affiliations:** 1Department of Dermatology, Health Management Center, Center for Plastic & Reconstructive Surgery, Zhejiang Provincial People’s Hospital (Affiliated People’s Hospital, Hangzhou Medical College), Hangzhou 310014, China; 2Graduate School of Clinical Medicine, Bengbu Medical College, Bengbu 233030, China; 3Department of Orthopedics, Xiangya Hospital, Central South University, Changsha 410000, China

**Keywords:** Nocardia, erythema nodosum, dermatology

## Abstract

Cutaneous nocardiosis is a rare bacterial infection that can result in various dermatologic manifestations such as actinomycetoma, lymphocutaneous infection, superficial skin infection, and secondary infection due to hematogenous dissemination. We report on a Chinese patient with erythema nodosum-like exanthema, possibly secondary to nocardiosis. Our diagnosis for this patient was based on the clinical presentation, histopathological evidence, and microbiological findings. Given the protean manifestation of Nocardia, persistent reports on new presentations of the disease are important for early identification and treatment.

## 1. Introduction

Erythema nodosum (EN) is a kind of panniculitis characterized by tender erythematous nodules, which affects the subcutaneous layer of the skin [1]. Currently, the diagnosis of EN relies predominantly on clinical presentation and pathological examination. Furthermore, EN is associated with many systemic diseases caused by multiple factors, including infections, neoplasia, inflammatory bowel diseases, and pregnancy [2,3].

## 2. Case Presentation

A 67-year-old man with a history of arterial hypertension was admitted to our hospital due to a painful erythema on the left forearm (Figure 1A) and erythematous swelling on both limbs (Figure 1B). He recalled being pricked in the left hand by a shrimp shell five months ago. The patient received cefotaxime treatment and underwent abscess drainage of the left forearm one month before the visit to our hospital; however, soon after the drainage, he experienced a fistula formation. One week ago, the patient experienced chills and fatigue, and his legs developed a few edematous erythema. There was obvious pressing pain in the erythema area. After admission to the hospital, the patient was empirically treated with levofloxacin (300 mg) once daily and thalidomide (50 mg) three times a day.

During his physical examination, a large edematous plaque with a fistula infection and fistula secretion on the left arm were noticed. Additionally, we found scattered edematous erythema on his lower limbs and a palpable node on the proximal portion of his thigh, which are suggestive of EN. However, blood tests showed a slight elevation in levels of C-reactive protein (44.5 mg/L) and the sedimentation rate of the erythrocytes (79 mm/first hour), routine blood, and serum biochemical parameters were normal. The results of tuberculin skin, syphilis, and HIV tests were all negative.

Skin tissue for a biopsy was taken from the calf. The biopsy of rashes on the lower limbs showed signs of panniculitis and erythrocyte extravasation in the adipose tissue. The interlobular septum edema revealed a dense infiltration of inflammatory cells consisting mainly of lymphocytes (Figure 2). Although the rashes on the lower limbs were suggestive of EN, the tissue culture tests for bacteria and fungi on the upper arm granuloma were both negative. Thus, we extracted grayish–white purulent fluid from his left arm with a needle and cultured the bacteria. A large number of filamentous Gram-positive bacteria with right angle branches were observed. We suspected that these bacteria were Nocardia. Hence, weak acid-fast staining was performed, and weak acid-positive bacteria were observed. The culture showed dry, chalky, white colonies in rabbit blood agar (Figure 3). Based on the clinical and microbiological findings, the patient was finally diagnosed with cutaneous nocardiosis and EN. Subsequently, he received a compound sulfamethoxazole tablet treatment (1.98 g, once/day). His condition significantly improved five days after the treatment. There was a reduction in the amount of purulent discharge from the fistulas, the rashes on both lower limbs subsided, and the pain disappeared. Upon discharge, the patient continued to receive oral TMP/SMX therapy for one month. The rashes on his left arm dried and scabbed over; the swelling in the left elbow receded; and the purulent secretion decreased. Despite his refusal of our recommendation for prolonged treatment, we kept following up with the patient to monitor his long-term outcomes. The rashes had mostly subsided at the four-month follow-up. There has been no recurrence of the skin rash on either of the lower limbs (Figure 4). Additionally, the infection has not recurred over the past nine years.

## 3. Discussion

The genus Nocardia is a branching filamentous Gram-positive bacillus [4]. This opportunistic bacillus is prevalent in soil, saprophytes, and air, and develops slowly in most patients [5]. Cutaneous nocardiosis is an uncommon Nocardia-induced cutaneous infection, which can be either primary or secondary. The clearance of Nocardia depends on various immune cells such as CD4 T cells, macrophages, and B cells [6]. Therefore, nocardiosis mainly occurs in immunocompromised patients. The risk factors include impaired local pulmonary defenses, long-term use of corticosteroids, lymphoreticular neoplasms, AIDS, and other immunosuppression-related conditions. An increase in Nocardia infections in immunocompetent patients has been seen in recent years. Beaman et al. [7] retrospectively analyzed 1000 nocardiosis cases and found that 62% of cases occurred in patients with susceptibility factors, while the remaining 38% of cases occurred in normal people. Some experts believe that these seemingly immunocompetent individuals may have some unrecognized primary immune defects or innate immune errors. However, our patient did not have any of these risk factors based on their past medical history.

Cutaneous nocardiosis can manifest in three different forms in immunocompetent individuals, including actinomycetoma, superficial skin infection, and lymphocutaneous infection. We searched the PubMed database for case reports and case series studies on cutaneous nocardiosis in immunocompetent patients published between 2012 and 2022. Seven cases were retrieved [8,9,10,11,12,13,14]. Primary cutaneous nocardiosis caused by Nocardia is rare in immunocompetent patients. Its main clinical manifestations include ulceration, pyoderma, cellulitis, nodules, and subcutaneous abscesses (Table 1). The clinical symptoms of our patient were consistent with previous cases, and a large edematous plaque with a fistula infection and fistula secretion were noticed on the left arm. In contrast to the localized lesions observed in immunocompetent patients, those with compromised immune function may have systemic symptoms, or even die [15,16]. Although Nocardia includes multiple bacteria species, there was no difference in the severity of the infection caused by these Nocardia species. According to previous reports, a history of trauma is the leading risk factor for the disease, and this disease mainly occurs in elderly people. Our patient is a 67-year-old man, and his lesion developed from a cut caused by shrimp shell.

The selection of an optimal treatment strategy for cutaneous nocardiosis relies on the severity and location of the infection, Nocardia species, host immunity, potential drug interactions, and toxicity of antimicrobial agents. Nocardia can be treated in many ways, including the incision and drainage of abscesses, the excision of lesions, antibiotic anti-infection, and the improvement of immune function. Nowadays, trimethoprim-sulfamethoxazole (TMP-SMX) is commonly used as the basic medicine for treating Nocardia [17]. Nevertheless, many studies suggest that combined drug therapy may have advantages compared to single-drug treatment, affording better clinical improvement, shorter treatment periods, and better prevention of recurrence [18]. Combined treatment drugs include amikacin, carbapenems, ceftriaxone, linezolid, minocycline, moxifloxacin, and levofloxacin [17,19]. The course of treatment varies from 7 days to 2 years according to the severity of the infection, clinical symptoms and immunosuppressive conditions [20,21]. Traditional intensive antibiotic therapy has been reported to increase survival rates, but the overall prognosis of nocardiosis patients remains relatively poor; additionally, coadministration of interferon gamma significantly improves the condition of those with serious infections [22]. Considering that our patient was elderly and that our hospital cannot conduct drug sensitivity testing, he received compound sulfamethoxazole tablet treatment. The skin lesions of the patient were obviously improved after the treatment.

EN is predominantly caused by infection, especially streptococcal infections and primary tuberculosis [2]. Other risk factors include drugs and vaccines, malignancies, and inflammatory bowel disease [3,23,24]. According to the detailed medical history that the patient reported, we could rule out these non-infectious factors. Although EN is more likely to correlate with other infectious aetiologies, the recent Nocardia infection may have been the potential trigger of EN in our patient. Primary cutaneous N. panniculitis infection has been described in two previous cases. In 1989, two cases of Nocardia infection showed the histopathologic finding of panniculitis [25].

The treatment of EN should focus mainly on the underlying cause of the condition [26]. It is common to use non-steroidal anti-inflammatory drugs (NSAIDs) to relieve pain, and systemic steroids may be used in severe cases. However, an underlying infection or malignancy should be ruled out before proceeding with corticosteroids [27]. In our study, the patient’s pain in his lower extremities improved significantly after five days of primary infection treatment and continued to improve during the follow-up period.

The Nocardia rubra cell-wall skeleton (Nr-CWS) promotes CD4 + T-cell activation and induces the secretion of TNF-α [28]. High levels of TNF-α have been reported in EN patients, possibly because it may make patients more susceptible to EN in the case of immune dysregulation, thus resulting in an excessive inflammatory response to bacteria such as Nocardia [29].

The clinical manifestations of cutaneous nocardiosis are nonspecific: mainly severe suppurative infection, cellulitis, and granulomatous changes. However, clinicians should be aware of its multiple clinical manifestations, including EN. In addition, since determining the underlying cause is critical for diagnosing and treating EN, dermatologists should identify Nocardia infections as early as possible to develop an optimal treatment strategy. In conclusion, this report describes a rare case of Nocardia infection, with EN and severe purulent infection as the primary symptoms.

## Figures and Tables

**Figure 1 medicina-58-01873-f001:**
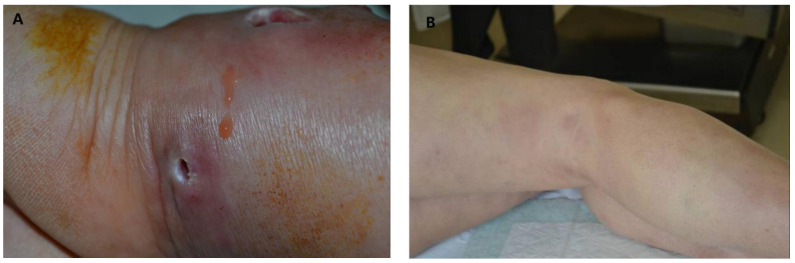
(**A**) Edematous plaques and fistulas on the left forearm (**B**) Erythematous nodules on both legs.

**Figure 2 medicina-58-01873-f002:**
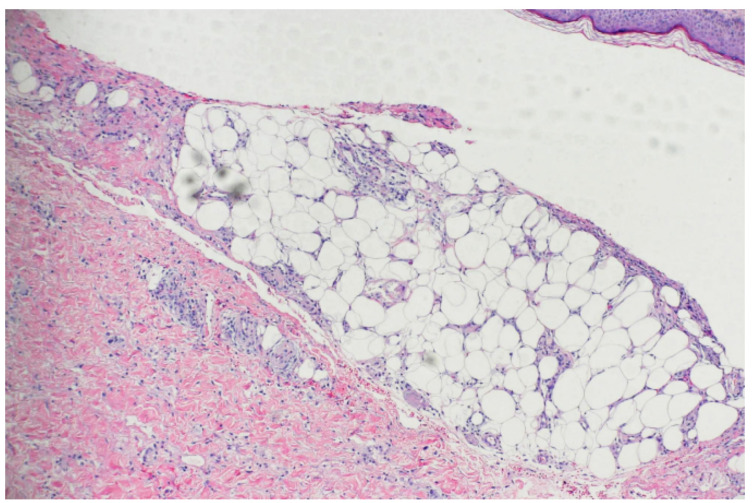
A representative pathological finding of erythema nodosum.

**Figure 3 medicina-58-01873-f003:**
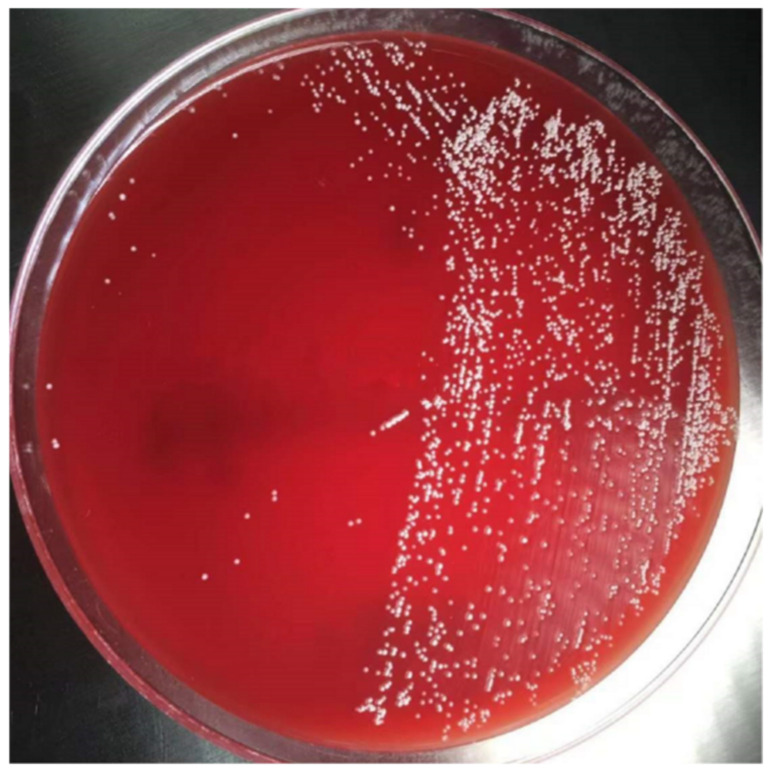
Bacterial colonies on rabbit blood agar after incubation at 37 °C for 5 days.

**Figure 4 medicina-58-01873-f004:**
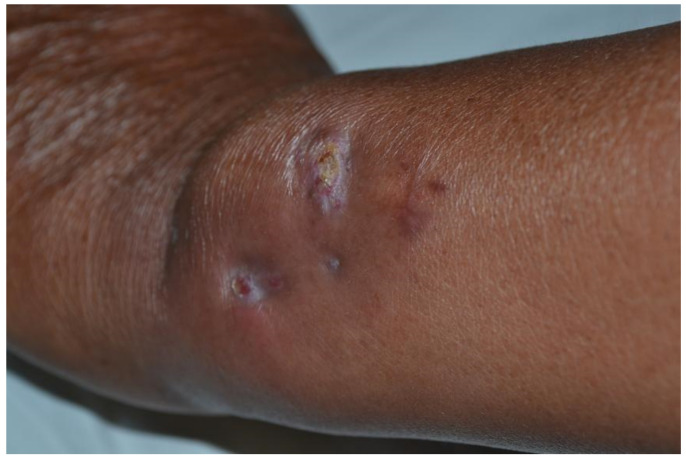
The patient achieved complete remission of the disease at the four-month follow-up.

**Table 1 medicina-58-01873-t001:** Clinical characteristics of seven cases of Nocardia infection.

Case/Ref	Age	Sex	Immunization Status	Clinical Symptoms	Sites of Infection	Treatment (Duration)	Outcome
1 [8]	50	W	Normal immune function	Intermittent fever at a temperature of 37.6 °C; a painful swelling of her face and neck on the right side; and a 3.5 × 3.5 cm diameter ulcerated plaque over the right temple.	Face and neck	3 months	Successful
2 [9]	42	W	Normal immune function	A painful palpable mass under her chin.	Chin	3 months	Successful
3 [10]	67	W	Normal immune function	Tender ulcers on the middle phalanx of her right middle finger with tender nodular lymphangitis extending up her forearm.	Right middle finger	3 months	Successful
4 [11]	83	W	Normal immune function	Erythema, purulent discharges, and edema of the infected wound.	Left leg	3 months	Successful
5 [12]	45	W	Normal immune function	Intermittent fever with a maximum temperature of 38.6 °C; a black–purplish lesion surrounded by an erythematous halo on the right leg.	Right leg	5 months	Successful
6 [13]	64	M	Normal immune function	A single 6 × 4 cm subcutaneous lesion with multiple scalp fistulas was observed in the posterior neck. There was a right purulent otorrhea coupled with pus discharge from the tumefaction and the fistulas.	Neck	18 months	Successful
7 [14]	85	W	Normal immune function	1.5 cm in diameter, with erythema margin, purulent discharge and local hyperthermia.	Dorsum of the right hand	1 month	Successful

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
