# Peer review of "Erythema Nodosum following Nocardia Infection: A Case Report"

_medicina, 2022, doi:10.3390/medicina58121873_

Round 1
Reviewer 1 Report (Previous Reviewer 1)
In this and other similar clinical cases it is important to do some serology to look for anti Nocardia IgG antibodies. There is an ELISA test available for that purpose that helps to monitor the response to therapy. A serological test is by far less expensive then Molecular techniques.
Author Response
Please see the attachment.

Reviewer 2 Report (Previous Reviewer 2)
The manuscript has been improved significantly and the authors corrected most of the previously mentioned issues. However, extensive English edition of language is still required, without this, manuscript can not be deemed for publication.
Author Response
Please see the attachment.

Reviewer 3 Report (New Reviewer)
This is an interesting case report about the patient with likely cutaneous nocardiosis who in addition to typical presentation developed erythema nodosum. This report adds to the current knowledge on this topic and might be worthwhile publishing, however few major addition/changes needs to be carried out before it can be considered.
1. Introduction line 29- inflammatory bowel diseases should be included here as well ( https://www.mdpi.com/2077-0383/10/2/364/htm)
2. case presentation- line 52" consistent with EN" - this should be changed to suggestive of.
3. just bacterial colonies on blood agar would not be enough to make diagnosis of nocardia spp; do you have any proof from more advanced molecular diagnostics? Do you have nocardia spp?
4. Line 87- it should be added that nocardia can affect not only immunocompromised host but immunocompetent as well. More should be elaborated on the immune status of your patient ( see the following : https://pubmed.ncbi.nlm.nih.gov/35454327/ AND https://www.mdpi.com/2076-2607/10/6/1120/htm
5. Please report if TB was ruled out? How about actinomycosis?
6. Discussion line 90- reference is missing ( https://www.mdpi.com/2077-0383/10/2/364/htm)
7. Discussion is very short. A detailed paragraph in regard of treatment of nocardiosis would be important.
8. Discussion- have you found any other similar cases? This should be discussed
9. references must be updated
Round 2
Reviewer 3 Report (New Reviewer)
I am pleased with the changes that authors have made. However, I was expecting more to see in the discussion section. Please compare clinical presentation of your cases with the other cases from the literature. For example, clinical presentation in immunocompromised and immunocompetent host are different and this should be clearly stated (https://pubmed.ncbi.nlm.nih.gov/35454327/)
Author Response
Please see the attachment.

This manuscript is a resubmission of an earlier submission. The following is a list of the peer review reports and author responses from that submission.
Round 1
Reviewer 1 Report
Identify the Nocardia specie that was isolated from patient. Morphology and growing characteristics are not dufficient to confirm etiological species.
Correct three minor spellings or meanings.
Reviewer 2 Report
Please see the attached file.

Reviewer 3 Report
In the biopsy that the authors mention as EN, the picture is small and no representative. It is suggested to show a better image, because is posible corresponded to the infection. The Nocardia specie was not identified